# The effect of birth weight on body composition: Evidence from a birth cohort and a Mendelian randomization study

Junxi Liu[1], Shiu Lun Au Yeung [1], Baoting He[1], Man Ki Kwok[1], Gabriel Matthew Leung[1], C. Mary Schooling [1,2]*

**1** School of Public Health, Li Ka Shing Faculty of Medicine, The University of Hong Kong, Hong Kong, China, **2** City University of New York Graduate School of Public Health and Health Policy, New York, New York, United States of America

* cms1@hku.hk

## Abstract

### Background

Lower birth weight is associated with diabetes although the underlying mechanisms are unclear. Muscle mass could be a modifiable link and hence a target of intervention. We assessed the associations of birth weight with muscle and fat mass observationally in a population with little socio-economic patterning of birth weight and using Mendelian randomization (MR) for validation.

### Methods

In the population-representative "Children of 1997" birth cohort (n = 8,327), we used multi-variable linear regression to assess the adjusted associations of birth weight (kg) with muscle mass (kg) and body fat (%) at ~17.5 years. Genetically predicted birth weight (effect size) was applied to summary genetic associations with fat-free mass and fat mass (kg) from the UK Biobank (n = ~331,000) to obtain unconfounded estimates using inverse-variance weighting.

### Results

Observationally, birth weight was positively associated with muscle mass (3.29 kg per kg birth weight, 95% confidence interval (CI) 2.83 to 3.75) and body fat (1.09% per kg birth weight, 95% CI 0.54 to 1.65). Stronger associations with muscle mass were observed in boys than in girls (p for interaction 0.004). Using MR, birth weight was positively associated with fat-free mass (0.77 kg per birth weight z-score, 95% CI 0.22 to 1.33) and fat mass (0.58, 95% CI 0.01 to 1.15). No difference by sex was evident.

### Conclusion

Higher birth weight increasing muscle mass may be relevant to lower birth weight increasing the risk of diabetes and suggests post-natal muscle mass as a potential target of intervention.

**Data Availability Statement:** Data are available upon request from the "Children of 1997" data access committee: aprmay97@hku.hk. The volume and complexity of the data collected preclude

public data deposition, because the participants could be identifiable from such extensive data which would comprise participant privacy. Data of the MR study are publicly available summary data.

**Funding:** This work is a substudy of the "Children of 1997" birth cohort which was initially supported by the Health Care and Promotion Fund, Health and Welfare Bureau, Government of the Hong Kong SAR [HCPF grant 216106] and reestablished in 2005 with support from the Health and Health Services Research Fund, Government of the Hong Kong SAR [HHSRF grant 03040771]; the Research Fund for the Control of Infectious Diseases in Hong Kong, the Government of Hong Kong SAR [RFCID grant 04050172]; the University Research Committee Strategic Research Theme (SRT) of Public Health, the University of Hong Kong. The Biobank clinical follow-up was partly supported by the WYNG Foundation. The funders had no role in study design, data collection and analysis, decision to publish, or preparation of the manuscript.

**Competing interests:** The authors have declared that no competing interests exist.

## Introduction

Observationally, lower birth weight is associated with higher risk of many chronic diseases including cardiovascular disease, diabetes and poor liver function,[1–4] but is also associated with lower risk of hormone-related cancers including breast and prostate cancer.[5, 6] Although these observations are open to confounding by factors such as socio-economic position (SEP), different associations by diseases suggest some of these associations may be causal. Mendelian randomization (MR) studies, taking advantages of the random allocation of genetic endowment at conception to obtain un-confounded estimates,[7] suggest an inverse association of birth weight with diabetes,[3, 4] but practical implications for prevention are unclear given birth weight is a complex phenotype. Elucidating the pathways linking birth weight with diabetes may provide additional insights into the identification of intervention targets, since birth weight is difficult to change[8] and does not have an "optimal" definition.[9]

Observationally, birth weight is positively associated with muscle mass in both teenagers and adults.[10, 11] Randomized controlled trials shows resistance training increases muscle mass and improves Hemoglobin A1c.[12] As such, muscle mass could be a modifiable downstream effect of birth weight, partially driven by sex hormones,[13, 14] potentially with sex-specific effects, consistent with the associations of lower birth weight with lower risk of breast and prostate cancers.[5, 6] However, previous observational studies assessing the role of birth weight in muscle mass sometimes adjusted for factors on the causal pathway, such as body mass index (BMI), height and physical activity, but may not fully adjusted for SEP.[15, 16]

To clarify the role of birth weight in body composition, we conducted two analyses with different assumptions and study designs (Fig 1). First, in an observational setting, we prospectively assessed the overall and sex-specific associations of birth weight with body composition (muscle mass, grip strength, and fat percentage) in a unique population, Hong Kong's "Children of 1997" birth cohort. In Hong Kong, the usual associations of higher SEP with higher birth weight and greater gestational age are almost absent,[17] and obesity has little socio-economic patterning in young people.[18] Therefore, Hong Kong is an ideal setting to assess the associations of birth weight and gestational age with body composition. We also assessed whether these associations differed by sex given the sex-difference in body composition since such differences are likely interpretable even when associations are confounded.[19] Second, using an MR design, we validated our findings, by assessing the associations of birth weight predicted by maternal genetics independent of fetal genetics, as a proxy of maternal intrauterine environment,[20] on body composition (fat-free mass, grip strength, and fat mass) in the largest publicly available genome wide association study (GWAS).[21] Taking advantage of the random allocation of genetic endowment at conception, MR studies provide un-confounded estimates and give the result of a lifelong difference in the risk factor between groups.[7]

## Material and methods

### Ethics statement

Ethical approval for the study, including comprehensive health related analyses, was obtained from Institutional Review Board of the University of Hong Kong/Hospital Authority Hong Kong West Cluster (HKU/HA HKW IRB). Informed written consent was obtained from the parents/guardians, or from the participant if 18 years or older, before participation in the Biobank Clinical Follow-up.

The MR study only uses published or publicly-available data. No original data were collected for the MR study. Ethical approval for each of the studies included in the investigation can be found in the original publications (including informed consent from each participant).

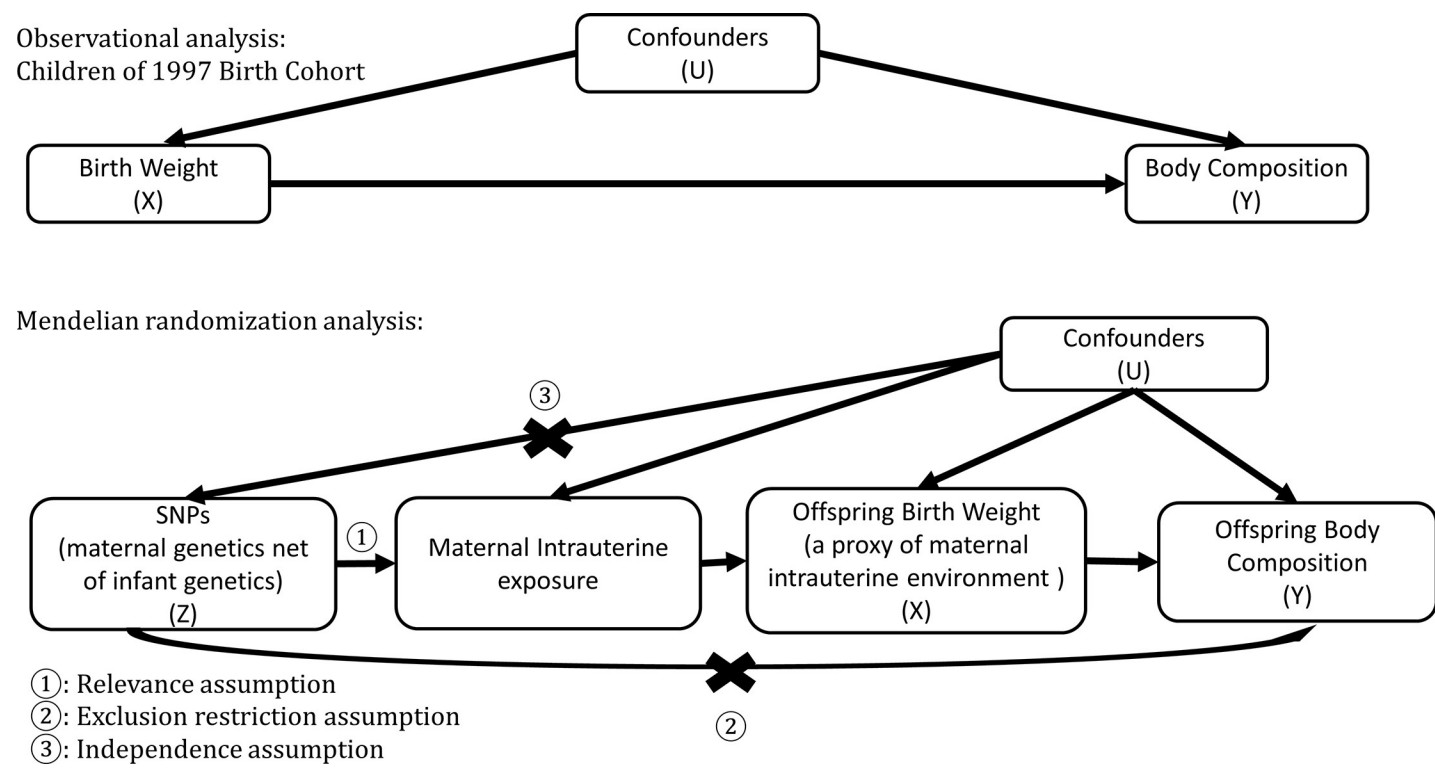

**Fig 1. Directed acyclic graph of the observational analysis and the Mendelian randomization analysis.**

## Observational study—The "Children of 1997" birth cohort

"Children of 1997" is a population-representative Chinese birth cohort (n = 8327), based on 88% of births in Hong Kong in April and May 1997.[22] The original study was designed to assess the associations of second-hand smoke exposure and breastfeeding with health services utilization in the first 18 months of life. Recruitment took place at all Maternal and Child Health Centers (MCHCs) in Hong Kong. Parents are strongly encouraged to take their children to the MCHCs for free preventive care and vaccinations to age 5 years. Parental and infant characteristics were obtained at recruitment. Contact was re-established in 2007. A Biobank clinical follow-up was conducted from 2013–2016 at ~17.5 years, when body composition was assessed from bio-impedance analysis using a Tanita segmental body composition monitor (Tanita BC-545, Tanita Co., Tokyo, Japan). Grip strength was measured using a Takei T.K.K.5401 GRIP D handgrip dynamometer (Takei Scientific Instruments Co. Ltd, Tokyo, Japan).

**Exposure—Birth weight, gestational age-specific birth weight z-score, and gestational age.** Birth weight recorded in grams was considered in kilograms and as internally generated gestational age-specific birth weight z-scores. Gestational age recorded in days was considered in weeks. Gestational age was calculated from the actual and expected dates of delivery reported by the mothers or primary caregivers at the initial MCHCs visit. The reported expected date of delivery is based on the date of the last menstrual period and any dating scans.

**Outcome—Body composition.** Muscle was assessed from whole-body muscle mass (kg), and dominant hand grip strength (kg). Fat mass was assessed from body fat percentage.

## Mendelian randomization study

**Exposure—Genetic predictors of maternal only effects on birth weight.** Single nucleotide polymorphisms (SNPs) predicting maternal effects on birth weight independent of fetal genetics (z-score transformed) at genome-wide significance (p-value$<5\times10^{-8}$) adjusted for gestational age where available (only available in $<$15% of the sample) and study-specific covariates were obtained from a GWAS consisting of two components, the Early Growth Genetics (EGG) Consortium (n = 12,319, 10 studies in the EGG consortium of European descent imputed up to the HapMap 2 reference panel, and n = 7,542, 2 studies in of European descent imputed up to the HRC panel) and the UK Biobank (n = 190,406, white European). A structural equation model was used to decompose the contributions of maternal genetic and fetal effects on birth weight (264,498 individuals own birth weight and 179,360 individuals offspring birth weight).[20]

We obtained independent SNPs ($R^2>0.01$) with the lowest p-value using the "*Clumping*" function of the MR-Base (*TwoSampleMR*) R package, with the 1000 Genomes catalog.[23] Potentially pleiotropic effects of these SNPs were obtained from up-to-date genotype to phenotype cross-references, i.e., GWAS Catalog (https://www.ebi.ac.uk/gwas/), Ensembl (http://www.ensembl.org/index.html) and Phenoscanner (http://www.phenoscanner.medschl.cam.ac.uk/). We also checked for potential pleiotropic effects and confounding of these SNPs from the Bonferroni corrected significance (12 traits $\times$ 30 SNPs, p-value$<1\times10^{-4}$) of their associations with alcohol consumption (past and current), smoking (past and current), physical activity (light, moderate, and vigorous), socioeconomic position (income and education), age of voice braking, age of menarche, and height in the UK Biobank summary statistics.[21]

**Outcome—Genetic associations with body composition.** Genetic associations with fat-free mass (kg), grip strength (kg) (left and right hand), and fat mass (kg) were obtained from the UK Biobank (~331,000 people of genetically verified white British ancestry). The genetic associations were assessed from multivariable linear regression adjusted for the first 20 principal components, sex, age, age-squared, the sex and age interaction and the sex and age-squared interaction.[21]

## Statistical analyses

**Observational analyses.** We compared "Children of 1997" who were included and excluded on baseline characteristics using chi-squared tests, and Cohen effect sizes[24] to obtain the magnitude of the differences between groups. Cohen effect sizes are usually categorized as 0.20 for small, 0.50 for medium and 0.80 for large for continuous variables, and as 0.10 for small, 0.30 for medium and 0.50 for large for categorical variables.

The associations of muscle mass, grip strength and fat percentage with potential confounders were assessed using independent t-tests or analysis of variance for continuous variables and chi-square tests for categorical variables. We used multivariable linear regression to obtain the observational associations of birth weight, birth weight z-score and gestational age with body composition adjusting for second-hand and maternal smoking, parental education, parental occupation, household income, type of housing, and sex. We additionally adjusted for gestational age in the association of birth weight with body composition. Sex differences were assessed from the significance of interaction terms adjusted for the other potential confounding interactions with sex.

Taking missingness into account, multiple imputation and inverse probability weighting were applied.[25] Firstly, we created 20 sets of imputed data accounting for missing confounders and exposures for all participants. Secondly, logistic regression was used to predict loss-to-follow-up based on gestational age (log-transformed because of the long tail of the

distribution), second-hand and maternal smoking, sex, type of housing, type of hospital at delivery, maternal migrant status, maternal age, and, breastfeeding with the lowest Akaike information criterion value. We also used the Hosmer-Lemeshow test to check model fit. Additionally, weights were checked to ensure acceptable stability. Unstable weights indicate model misspecification.[25] Lastly, we combined each inverse probability weighting effect estimator and its corresponding sandwich variance estimator according to Rubin's Rules.[26]

**Mendelian randomization.** The strength of the genetic instruments was assessed from the *F*-statistic, obtained using an approximation (square of SNP on exposure divided by variance of SNP on exposure).[27, 28] A higher *F*-statistic indicates a lower risk of weak instrument bias.[27] The effects of birth weight on the outcomes were obtained from a meta-analysis of SNP-specific Wald estimates (SNP-outcome association divided by SNP-exposure association) using inverse variance weighting with multiplicative random effects assuming balanced pleiotropy. Heterogeneity of the Wald estimates was assessed from the $I^2$ statistic, where a high $I^2$ may indicate the presence of invalid SNPs.[29] Differences by sex were additionally assessed.[30] Power calculations were performed using the approximation that the sample size for Mendelian randomization equates to that of the same regression analysis with the sample size divided by the $r^2$ for genetic variant on exposure.[31]

**Sensitivity analyses relevant to the observational designs.** A complete case analysis was conducted as a validation without taking missingness into account.

**Sensitivity analyses relevant to Mendelian randomization.** As sensitivity analyses, we excluded SNPs which may be invalid. These included 1)SNPs associated with potentially pleiotropic effects on muscle or fat given in Ensembl or the GWAS Catalog; 2) SNPs associated with potential confounders and/or pleiotropic effects in the UK Biobank at Bonferroni corrected significance (p-value$<1\times10^{-4}$) and in PhenoScanner (p-value $<1\times10^{-5}$).

Estimates were obtained from sensitivity analyses with different assumptions.

Specifically, we used a weighted median which may generate correct estimates if >50% of weight is contributed by valid SNPs.[32] MR-Egger was used which generates correct estimates if all the SNPs are invalid instruments as long the instrument strength independent of direct effect assumption is satisfied.[29] A non-null intercept from MR-Egger indicates potential directional pleiotropy and an invalid inverse variance weighting estimate.[32] The Mendelian randomization pleiotropy residual sum and outlier (MR-PRESSO) was additionally used, which detects and corrects for pleiotropic outliers assuming >50% of the instruments are valid, balanced pleiotropy and the instrument strength independent of direct effect assumption are satisfied.[33, 34]

All statistical analyses were conducted using R version 3.4.2 (R Foundation for Statistical Computing, Vienna, Austria). The R packages *MendelianRandomization* [35] and *MRPRESSO* [34] were used to generate the estimates.

## Results

### Children of 1997

Among the originally recruited 8327 participants, 6850 are contactable and living in Hong Kong. 3460 (51%) participated in the Biobank clinical follow-up, of which 3455 had muscle mass, grip strength or fat percentage (Fig 2). The mean and standard deviation (SD) of muscle mass, grip strength and fat percentage were 42.6kg (SD 8.8kg), 25.8kg (SD 8.3kg) and 21.7% (SD 8.8%). Boys had higher muscle mass and grip strength but lower fat percentage than girls. Body composition had little association with SEP (Table 1). Differences between participants included and excluded from the study were found for gestational age, sex, second-hand and

maternal smoking exposure, and SEP using chi-squared tests, but the magnitude of these differences was small (Cohen effect size <0.15) (S1 Table).

Observationally, birth weight and birth weight z-score were positively associated with muscle mass, grip strength, and, fat percentage. The associations were strengthened after adjusting for gestational age. Gestational age was not associated with muscle mass, grip strength or fat percentage. Associations with muscle muss differed by sex for birth weight z-score and birth weight adjusted for gestational age, with stronger associations in boys (Table 2). Similar estimates were obtained in the complete case analyses (S2 Table).

### Mendelian randomization

**Genetic instruments for maternal only effects on birth weight.** Altogether, 30 SNPs independently predicted effects of maternal genetics net of infant genetics on birth weight (p-value<$5 \times 10^{-8}$) in people of European ancestry.[20] The average of SNP-specific $F$ statistics was 79, and all were >30 (S3 Table); the variance explained ($r^2$) was 0.013. As such, the MR study had 80% power with 5% alpha to detect a difference of 0.04 of an effect size in fat-free mass and fat mass per z-score of birth weight.

Of the 30 SNPs predicting birth weight, 5 palindromic SNPs were aligned (S3 Table); 5 SNPs had potentially pleiotropic effects, i.e., (height and metabolic response) in Ensembl or the GWAS Catalog. Of the remaining 25 SNPs, 15 remained after excluding SNPs related to height, menarche, income, and basal metabolic rate in the UK Biobank (p-value<$1 \times 10^{-4}$) and in PhenoScanner (p-value <$1 \times 10^{-5}$) (S4 and S5 Tables).

**Mendelian randomization estimates.** Based on all 30 SNPs, genetically predicted birth weight (maternal effects net of infant effects) was positively associated with fat-free mass, fat mass, and grip strength. No sex differences were evident. After excluding 5 potentially pleiotropic SNPs, the positive associations remained, however, the associations were not robust after additionally excluding 10 potentially pleiotropic and confounded SNPs (S5 Table). Detecting and correcting for pleiotropic outliers, MR-PRESSO indicated robust positive estimates, in particular with fat mass (Fig 3). MR-Egger had wider confidence intervals but had no indication of potential pleiotropy (S5 Table).

## Discussion

Using two different designs, with different assumptions and data sources, we found consistent evidence that birth weight was positively associated with muscle mass (fat-free mass), grip strength and fat percentage (fat mass). These findings are consistent with previous observational studies,[10, 36, 37] but add by validating these observations in a setting with little socio-economic patterning of birth weight and the use of MR.

These two study designs have contrasting limitations. First, residual confounding could not be ruled out in the observational design. SEP is hard to measure precisely and eliminate. In Hong Kong, the usual positive association of SEP with birth weight and gestational age is almost absent,[17] and SEP has little association with adiposity in young people.[18] However, other familial factors might affect birth weight and body composition.[38, 39] It is also difficult to disentangle correlated factors reliably in an observational study. Second, follow-up was incomplete (51%). Selection bias is unlikely, given no major difference between the participants with and without body composition indices. Moreover, differences by sex were observed, which are less open to confounding.[19] Third, MR studies have stringent assumptions, i.e., the genetic instruments should strongly predict the exposure, should not be confounded and should only be linked with the outcomes via the exposure. To examine the robustness of our findings, we excluded SNPs which may have pleiotropic effects or be associated with potential

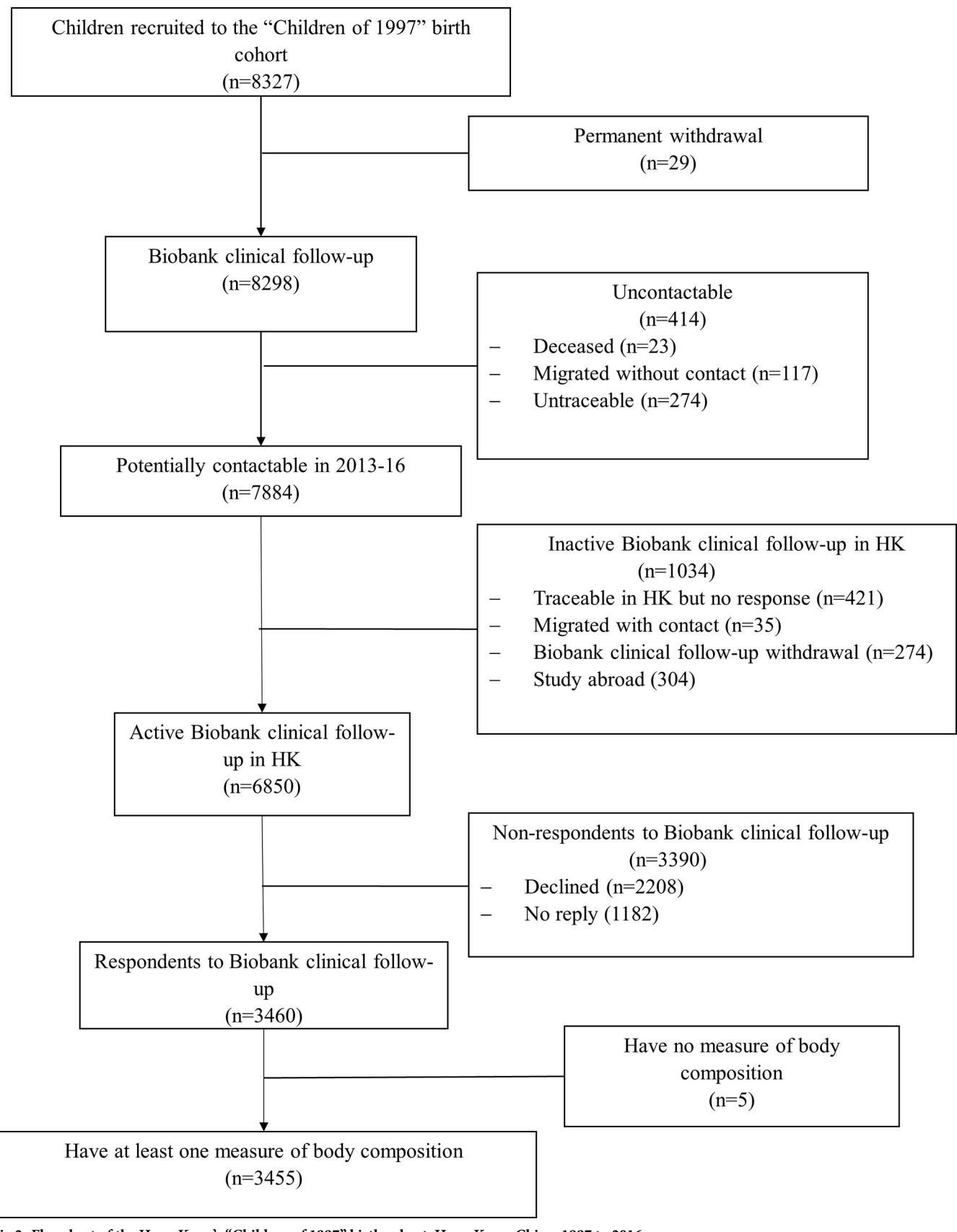

**Fig 2. Flowchart of the Hong Kong's "Children of 1997" birth cohort, Hong Kong, China, 1997 to 2016.**

confounders, and the results were similar. MR-PRESSO also gave consistently positive sex-specific estimates after taking potential pleiotropy into account (Fig 3). Although some of the $I^2$ were large, after excluding potentially pleiotropic and/or confounded SNPs, they became

**Table 1. Baseline characteristics muscle mass, grip strength, and fat percentage among participants in Hong Kong's "Children of 1997" birth cohort, Hong Kong, China, 1997 to 2016.**

| Characteristics | Muscle mass (kg) | | | | Grip strength (kg) | | | | Fat percentage (%) | | | |
|---|---|---|---|---|---|---|---|---|---|---|---|---|
| | No. | % | Mean (SD) | P-value[a] | No. | % | Mean (SD) | P-value[a] | No. | % | Mean (SD) | P-value[a] |
| Muscle mass (kg) | 3440 | | 42.6 (8.8) | | | | | | | | | |
| Grip strength (kg) | | | | | 3444 | | 25.8 (8.3) | | | | | |
| Fat percentage (%) | | | | | | | | | 3452 | | 21.7 (8.8) | |
| Sex | 3440 | | | <0.001 | 3444 | | | <0.001 | 3452 | | | <0.001 |
| Girl | 1707 | 49.6% | 35.3 (3.4) | | 1710 | 49.7% | 19.9 (4.5) | | 1714 | 49.7% | 28.1 (5.9) | |
| Boy | 1733 | 50.4% | 49.7 (6.3) | | 1734 | 50.3% | 31.6 (7.0) | | 1738 | 50.3% | 15.3 (6.4) | |
| Unknown | 0 | 0.0% | - | | 0 | 0.0% | - | | 0 | 0.0% | - | |
| Second-hand and maternal smoking exposure | 3440 | | | 0.07 | 3444 | | | 0.77 | 3452 | | | 0.17 |
| None | 940 | 27.3% | 42.1 (8.4) | | 939 | 27.3% | 25.6 (8.1) | | 943 | 27.3% | 21.2 (8.5) | |
| Prenatal second-hand smoking | 1275 | 37.1% | 42.7 (8.8) | | 1276 | 37.0% | 26.0 (8.4) | | 1276 | 37.0% | 21.6 (9.0) | |
| Postnatal second-hand smoking | 953 | 27.7% | 43.0 (9.2) | | 956 | 27.8% | 25.7 (8.3) | | 960 | 27.8% | 22.0 (9.0) | |
| Maternal smoking | 128 | 3.7% | 42.7 (8.8) | | 128 | 3.7% | 26.0 (8.2) | | 128 | 3.7% | 22.9 (8.6) | |
| Unknown | 144 | 4.2% | 41.1 (8.6) | | 145 | 4.2% | 25.3 (8.7) | | 145 | 4.2% | 21.9 (9.0) | |
| Highest parental education level | 3440 | | | 0.06 | 3444 | | | 0.12 | 3452 | | | 0.04 |
| Grade< = 9 | 984 | 28.6% | 42.2 (9.1) | | 988 | 28.7% | 25.4 (8.3) | | 989 | 28.7% | 22.2 (9.0) | |
| Grades 10–11 | 1481 | 43.1% | 42.4 (8.6) | | 1483 | 43.1% | 25.7 (8.4) | | 1488 | 43.1% | 21.6 (8.8) | |
| Grades> = 12 | 959 | 27.9% | 43.1 (8.9) | | 957 | 27.8% | 26.3 (8.1) | | 959 | 27.8% | 21.1 (8.7) | |
| Unknown | 16 | 0.5% | 39.7 (7.3) | | 16 | 0.5% | 24.4 (6.8) | | 16 | 0.5% | 23.9 (8.6) | |
| Highest parental occupation | 3440 | | | 0.32 | 3444 | | | 0.04 | 3452 | | | 0.12 |
| I(unskilled) | 98 | 2.8% | 41.9 (9.3) | | 99 | 2.9% | 25.4 (8.6) | | 99 | 2.9% | 21.8 (8.1) | |
| II(semiskilled) | 281 | 8.2% | 43.0 (9.0) | | 283 | 8.2% | 26.4 (8.3) | | 285 | 8.3% | 21.9 (8.8) | |
| III(semiskilled) | 503 | 14.6% | 42.3 (9.0) | | 504 | 14.6% | 25.1 (8.4) | | 503 | 14.6% | 21.5 (8.8) | |
| III(nonmanual skilled) | 876 | 25.5% | 42.4 (8.7) | | 878 | 25.5% | 25.4 (8.1) | | 879 | 25.5% | 22.2 (9.2) | |
| IV (managerial) | 438 | 12.7% | 43.2 (9.5) | | 438 | 12.7% | 26.5 (8.5) | | 439 | 12.7% | 22.2 (8.6) | |
| V(professional) | 794 | 23.1% | 42.8 (8.5) | | 792 | 23.0% | 26.2 (8.2) | | 795 | 23.0% | 21.0 (8.5) | |
| Unknown | 450 | 13.1% | 42.0 (8.5) | | 450 | 13.1% | 25.3 (8.4) | | 452 | 13.1% | 21.5 (9.2) | |
| Household income per head at recruitment | 3440 | | | 0.07 | 3444 | | | 0.16 | 3452 | | | 0.15 |
| First quintile | 566 | 16.5% | 42.0 (8.5) | | 572 | 16.6% | 25.6 (8.5) | | 571 | 16.5% | 21.7 (8.9) | |
| Second quintile | 613 | 17.8% | 41.9 (9.3) | | 613 | 17.8% | 25.0 (8.3) | | 616 | 17.8% | 22.2 (8.7) | |
| Third quintile | 616 | 17.9% | 43.3 (8.8) | | 617 | 17.9% | 26.1 (8.3) | | 618 | 17.9% | 21.8 (9.1) | |
| Fourth quintile | 630 | 18.3% | 42.7 (8.9) | | 629 | 18.3% | 25.9 (8.5) | | 630 | 18.3% | 21.2 (8.7) | |
| Fifth quintile | 644 | 18.7% | 42.9 (8.6) | | 642 | 18.6% | 26.1 (7.9) | | 645 | 18.7% | 21.1 (8.5) | |
| Unknown | 371 | 10.8% | 42.6 (9.0) | | 371 | 10.8% | 26.1 (8.3) | | 372 | 10.8% | 22.2 (9.2) | |
| Type of housing at recruitment | 3440 | | | 0.45 | 3444 | | | 0.44 | 3452 | | | 0.36 |
| Public | 1435 | 41.7% | 42.5 (8.9) | | 1440 | 41.8% | 25.8 (8.5) | | 1445 | 41.9% | 21.9 (9.1) | |
| Subsidized home ownership scheme | 545 | 15.8% | 42.2 (8.8) | | 541 | 15.7% | 25.2 (8.2) | | 544 | 15.8% | 22.0 (8.9) | |
| Private | 1355 | 39.4% | 42.8 (8.8) | | 1358 | 39.4% | 25.9 (8.1) | | 1358 | 39.3% | 21.3 (8.5) | |
| Unknown | 105 | 3.1% | 41.8 (8.8) | | 105 | 3.0% | 25.8 (8.7) | | 105 | 3.0% | 21.2 (8.7) | |

[a] Using independent t-test or analysis of variance for continuous variables and chi-square tests for categorical variables

**Table 2. Adjusted associations of birth weight, birth weight z-score and gestational age with body composition with inverse probability weighting (IPW) and multiple imputation (MI) in the Hong Kong's "Children of 1997" birth cohort, Hong Kong, China, 1997 to 2016.**

| Outcome | Exposure | Sex-adjusted as confounder | | p-value of interaction with sex | Boys | | Girls | |
|---|---|---|---|---|---|---|---|---|
| | | Beta | 95% CI | | Beta | 95% CI | Beta | 95% CI |
| Muscle mass (kg) | Birth weight (kg) | 2.32 | 1.94 to 2.70 | 0.12 | 2.59 | 1.95 to 3.23 | 1.99 | 1.61 to 2.36 |
| | Birth weight z-score | 1.29 | 1.12 to 1.47 | 0.002 | 1.54 | 1.25 to 1.83 | 1.01 | 0.84 to 1.17 |
| | Birth weight adjusted for gestational age | 3.29 | 2.83 to 3.75 | 0.004 | 3.89 | 3.12 to 4.66 | 2.58 | 2.14 to 3.03 |
| | Gestational age (week) | 0.00 | -0.10 to 0.10 | 0.25 | -0.06 | -0.23 to 0.12 | 0.07 | -0.04 to 0.17 |
| Grip strength (kg) | Birth weight (kg) | 1.39 | 0.95 to 1.84 | 0.36 | 1.58 | 0.86 to 2.30 | 1.16 | 0.66 to 1.66 |
| | Birth weight z-score | 0.68 | 0.48 to 0.89 | 0.35 | 0.77 | 0.44 to 1.10 | 0.57 | 0.35 to 0.80 |
| | Birth weight adjusted for gestational age | 1.75 | 1.22 to 2.29 | 0.29 | 2.01 | 1.14 to 2.87 | 1.43 | 0.83 to 2.02 |
| | Gestational age (week) | 0.08 | -0.04 to 0.20 | 0.89 | 0.09 | -0.11 to 0.28 | 0.07 | -0.06 to 0.21 |
| Fat percentage | Birth weight (kg) | 0.58 | 0.11 to 1.04 | 0.30 | 0.35 | -0.32 to 1.01 | 0.85 | 0.20 to 1.50 |
| | Birth weight z-score | 0.44 | 0.23 to 0.65 | 0.58 | 0.39 | 0.09 to 0.69 | 0.51 | 0.22 to 0.80 |
| | Birth weight adjusted for gestational age | 1.09 | 0.54 to 1.65 | 0.69 | 1.00 | 0.20 to 1.79 | 1.23 | 0.46 to 2.00 |
| | Gestational age (week) | -0.09 | -0.22 to 0.03 | 0.23 | -0.17 | -0.34 to 0.01 | -0.01 | -0.19 to 0.16 |

Adjustment: second-hand and maternal smoking, highest parental education, parental occupation, household income, type of housing and sex.

smaller. MR-Egger regression did not show directional pleiotropy even though the intercept test might be underpowered. Fourth, the overlap of the GWAS of birth weight with UK Biobank is ~90%, which might bias estimates towards the exposure-outcome association, nevertheless, the $F$ statistic was 79 suggesting weak instrument bias is less likely.[27] Fifth, the MR

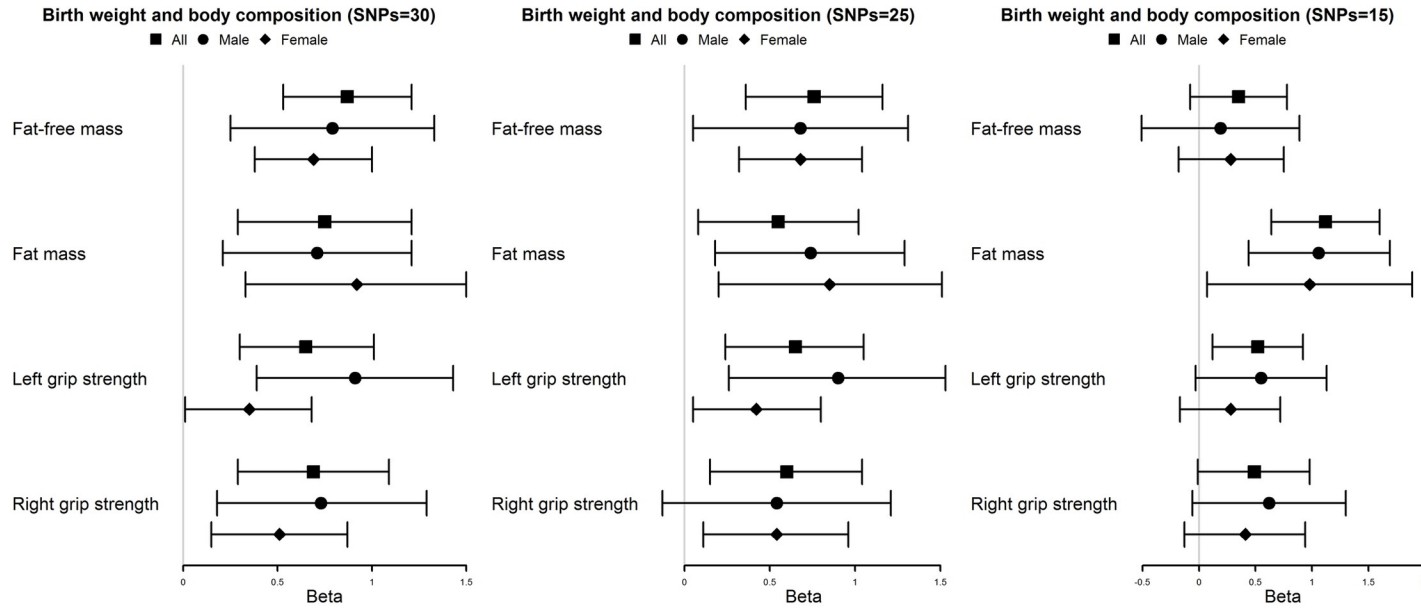

**Fig 3. Mendelian randomization estimates of the effect of genetically predicted birth weight (maternal effects net of infant effects) (per z-score) on body composition with and without potentially pleiotropic SNPs and potentially confounded SNPs using MR-PRESSO.** SNP = 30: all SNPs; SNP = 25, excluding maternal genotype related SNPs, and potential pleiotropic SNPs from GWAS catalog and Ensembl: rs560887 (*G6PC2*), rs2971669 (*GCK*), rs148982377 (*ZNF789*), rs2168101 (*LMO1*), rs10830963 (*MTNR1B*); excluding potential pleiotropic and/or confounded SNPs in UK Biobank in Bonferroni corrected significance (p-value<1×10$^{-4}$) and in PhenoScanner (p-value<1×10$^{-5}$): rs934232 (*ZFP36L2*), rs34471628 (*DUSP1*), rs9379084 (*RREB1*), rs6911024 (*MICA*), rs6995390 (*ZFHX4*), rs10814916 (*GLIS3*), rs111867185 (*AGBL2*), rs6487930 (*IPO8*), rs180438 (*SLC38A4*), rs597808 (*ATXN2*). MR-PRESSO: Mendelian randomization pleiotropy residual sum and outlier.

study mainly pertains to people of European ancestry. However, restricting the MR study to the European ancestry could mitigate the potential confounding bias caused by hidden population structure, if the genetic associations vary by ethnic groups.[40] Ethnic differences between the MR study and the observational study is another concern, although we usually expect causal factors to act consistently across populations, unless we have evidence that the causal mechanism differs or is less relevant in some specific populations. Given the distribution of body composition varies by ethnicity, it is possible that the drivers of body composition also vary by ethnicity. However, more parsimoniously, it is likely that the drivers of body composition are similar across populations but their relevance varies. However, causes are usually consistent although not relevant in all contexts. Replicating the MR study in a Chinese population would be very helpful. Sixth, using summary statistics from different samples in the MR study means differences by age and sex could not be comprehensively assessed since no sex-specific genetic predictors of birth weight are available and hence we were only able to assess differences by sex observationally. Seventh, canalization might compensate for genetic variation in birth weight. However, whether such canalization exists is unknown. Eighth, MR provides an estimate of the effect of lifetime exposure rather than indicating the exact size of the corresponding intervention, as such it indicates an etiological pathway. Birth weight is affected by maternal and fetal genetics.[20, 41, 42] We used maternal genetics predictors net of infant genetics so the associations found with offspring body composition indicate the role of the intrauterine environment. Whether the intrauterine environment is a modifiable target of intervention, or whether subsequent consequences of the intrauterine environment would be more suitable for intervention requires investigation. Lastly, different genetic effects by generation is a concern. Given summary data was used, the genetic effects of maternal genetics net of infant genetics with offspring body composition were approximated by the genetic effects of maternal genetics net of infant genetics with maternal body composition. However, effects of genetic are likely consistent across generations.[43] We cannot rule out the possibility of the gene-environmental and/or gene-gene interactions leading to heritable epigenetic changes, which requires further exploration with individual maternal and infant genetic data.[43]

Positive associations of birth weight with body composition seem intuitive and might arise for several reasons. Development before birth is critical for skeletal muscle and adiposity. Specifically, myogenesis forms most fiber, and muscle fiber numbers do not increase after birth. [36, 44] Similarly, fat cell number is complete at birth and postnatal fat mass is mainly via increasing adipocyte size.[45, 46] Mechanisms driving differential development of muscle and fat cells before birth are unclear, but likely related to nutrition, acting via hormones. We have previously proposed that lower levels of androgens might cause higher diabetes risks via lower muscle mass.[13, 14, 47] Lower birth weight might indicate lower levels of androgens thus generating positive associations of birth weight with muscle mass and the stronger associations in men seen in both the observational and MR designs, although a difference by sex was not evident in the MR design. From an etiological perspective, a causal association of birth weight with muscle mass provides a potential mechanistic, a modifiable pathway from lower birth weight to higher diabetes risks.[3, 4, 47] Given birth weight is difficult to change, such findings suggest that muscle building might reduce diabetes risk due to lower birth weight. Such a mechanism, might also help explain a higher risk of diabetes in Asia with low prevalence of obesity, lower birth weight, and lower muscle mass than in western settings.[48–52] Asians have more than double the risk of developing diabetes than Europeans at the same BMI.[48] However, it is possible that the observed associations do not extend to the extremes of the birth weight distribution, where birth weight may be a symptom of specific pathology. Given this is likely to be rare, we do not have sufficient sample size to assess this possibility. These findings are consistent with the idea of evolutionary public health, i.e., that the trade-off of

growth and reproduction against longevity may inform understanding of chronic diseases and the identification of interventions.

## Conclusion

Higher birth weight might increase fat-free mass and fat mass. Our study provides some indications that low fat-free mass may explain why lower birth weight increases diabetes risk and suggests muscle building as an attractive target of intervention.

## Supporting information

**S1 Table. Baseline characteristics of the participants who were included (n = 3455) and excluded (n = 4872) in the analyses of the Hong Kong's "Children of 1997" birth cohort, Hong Kong, China, 1997 to 2016.**
(DOCX)

**S2 Table. Adjusted associations of birth weight, birth weight z-score and gestational age with body composition in complete case analysis in the Hong Kong's "Children of 1997" birth cohort, Hong Kong, China, 1997 to 2016.**
(DOCX)

**S3 Table. Single nucleotide polymorphisms (SNPs) independently predicted effects of maternal genetics net of infant genetics on birth weight in Europeans from the Early Growth Genetics (EGG) Consortium (p-value$<5\times10^{-8}$).**
(DOCX)

**S4 Table. Single nucleotide polymorphisms (SNPs) with potential pleiotropic effects, and/ or potential confounders from Ensembl, GWAS Catalog, PhenoScanner, and UK Biobank.**
(DOCX)

**S5 Table. Estimates of the effect of genetically predicted birth weight (maternal effects net of infant effects) (per z-score) on body composition with and without potentially pleiotropic single nucleotide polymorphisms (SNPs) and potentially confounded SNPs using Mendelian randomization with different methodological approaches.**
(DOCX)

## Acknowledgments

The authors thank colleagues at the Student Health Service and Family Health Service of the Department of Health for their assistance and collaboration. They also thank late Dr. Connie O for coordinating the project and all the fieldwork for the initial study in 1997–1998.

## Author Contributions

**Conceptualization:** Shiu Lun Au Yeung, C. Mary Schooling.

**Data curation:** Man Ki Kwok.

**Formal analysis:** Junxi Liu.

**Investigation:** Junxi Liu, Baoting He.

**Methodology:** Shiu Lun Au Yeung, C. Mary Schooling.

**Supervision:** Shiu Lun Au Yeung, Gabriel Matthew Leung, C. Mary Schooling.

**Writing – original draft:** Junxi Liu.

**Writing – review & editing:** Junxi Liu, Shiu Lun Au Yeung, C. Mary Schooling.

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
