## [Decision Letter · Decision Letter 0]

21 Jul 2019

PONE-D-19-16847

The effect of birth weight on body composition: Evidence from a birth cohort and a Mendelian randomization study

PLOS ONE

Dear Dr Schooling,

Thank you for submitting your manuscript to PLOS ONE. After careful consideration, we feel that it has merit but does not fully meet PLOS ONE’s publication criteria as it currently stands. Therefore, we invite you to submit a revised version of the manuscript that addresses the points raised during the review process.

We would appreciate receiving your revised manuscript by Sep 04 2019 11:59PM. To enhance the reproducibility of your results, we recommend that if applicable you deposit your laboratory protocols in protocols.io, where a protocol can be assigned its own identifier (DOI) such that it can be cited independently in the future. For instructions see: http://journals.plos.org/plosone/s/submission-guidelines#loc-laboratory-protocols

We look forward to receiving your revised manuscript.

Kind regards,

David Meyre

Academic Editor

PLOS ONE

Journal Requirements:

Reviewers' comments:

Reviewer's Responses to Questions

**Comments to the Author**

1. Is the manuscript technically sound, and do the data support the conclusions?

Reviewer #1: Yes

Reviewer #2: Partly

2. Has the statistical analysis been performed appropriately and rigorously? 

Reviewer #1: Yes

Reviewer #2: No

3. Have the authors made all data underlying the findings in their manuscript fully available?

Reviewer #1: Yes

Reviewer #2: Yes

4. Is the manuscript presented in an intelligible fashion and written in standard English?

Reviewer #1: Yes

Reviewer #2: Yes

5. Review Comments to the Author

Reviewer #1: It was a pleasure to review this comprehensive study performed by Liu and colleagues, investigating the effect of birth weight on consequent body composition. The authors are experienced with the applied methods and have taken care in the performing appropriate analyses.

The work is already close to a standard suitable for publication. My main suggestion for improvement would be some clarification regarding the interpretation of causality. I think it is important that the authors discuss the potential implications of a shared (genetic) aetiology between birth weight and consequent body composition. Keeping this in mind, is it justified that the authors should imply in their conclusion that birth weight has a causal effect on consequent body composition?

In any case, I think that the authors should also be more clear about whether they are genuinely suggesting birth weight as a target for clinical intervention - although they do state this in the last line of the conclusion, I must admit that I'm struggling to take it seriously. The practicalities of modifying birth weight is another consideration. How might this be achieved? If the authors prefer not to answer this, then they should probably remove this suggestion.

Some discussion about the range of birth weights for which these findings might apply is also warranted. Can these estimates be extrapolated to the extremes of birth weight, for example?

Finally, I wonder whether the authors might also consider replacing (or supplementing) some of the existing results tables with figures (e.g. forest plots of the estimates and their confidence intervals). The current format used for presenting the data is a little overwhelming in places and therefore difficult to follow. Figures might make things easier to digest.

Reviewer #2: The authors examined the associations of birth weight (BW) with muscle and fat mass in an observational study. They also performed a Mendelian Randomisation (MR) of BW on these measures in the UK Biobank study. I have several major concerns relating to the MR section of the paper.

The authors do not clearly state the question that they are seeking to answer with the BW-fat/muscle MR analysis. It would be helpful if the authors could provide a diagram (eg. a DAG) explaining the causal question which they seek to answer. For example, are the authors using birth weight as a proxy for intrauterine exposures which might influence muscle and fat mass? If so, the maternal birth weight-associated alleles that are not passed to the child are the appropriate instrumental variables to use. The issue of MR in this context was discussed in the latest BW GWAS paper (https://doi.org/10.1038/s41588-019-0403-1). Further papers discussing this issue are https://doi.org/10.12688/wellcomeopenres.10567.1 and https://doi.org/10.1093/ije/dyz019. Alternatively, the authors might be asking about whether the genetic variation underlying birth weight is shared with the underlying muscle and fat mass, in which case, the fetal genotype is more important. Could the authors please clarify this.

A major problem in MR with birth weight using fetal genetic instruments as the exposure is the violation of MR assumptions by the association of maternal genotype with the outcome of interest. The latest BW GWAS paper discovered additional BW associated SNPs and also quantified the independent maternal and fetal associations with BW at each of the loci. Ideally this larger list and independent fetal effect sizes should be used for the exposure SNPs, the summary statistics from these analyses adjusted for maternal/fetal genotype associations are available to download from the EGG consortium website http://egg-consortium.org/birth-weight-2019.html. These adjusted association statistics correct the SNP-exposure associations for maternal genetic effects. This still would not get around the problem of the potential violation of the MR assumptions, indeed the large attenuation of the association results when potentially maternal SNPs are excluded from the analyses suggests that failing to account for maternal effects may be affecting the results. To properly perform these analyses the SNP-outcome associations should be corrected for maternal genetic associations. If this is not possible, this limitation should be clearly stated and fully discussed in the paper. The confounding introduced by maternal genotype is briefly mentioned at the end of the limitations section, but the potential impact of this on the results and the interpretation of them is not given sufficient attention and need to be carefully discussed and made more prominent.

It is also not clear to me why the authors reduced the original list of 60 BW SNPs to 47. The list of 60 loci already represents a list of independent association signals. A lot of the SNPs which were removed seem to be correlation artefacts rather than genuine LD, for example the SNP rs61830764 (DTL) is over 50mb from the nearest signal suggesting that the correlation here doesn’t reflect real LD. The signal rs1011939 (GPR139) also is the only SNP on chromosome 16 but seems to have been dropped as “not independent” although there can be no real LD with the other signals.

Other comments

In the limitations section, point 6 leaves me confused. They state “no age-specific genetic predictors of BW are available”, but birth weight is by definition the weight at birth, there is no age variation in BW. Point 8 also leaves me a little confused. It talks about lifetime exposure, but birth weight is again defined only at birth. I suggest the authors rephrase this point as I’m not entirely sure from what they have written what they are trying to say here.

6. PLOS authors have the option to publish the peer review history of their article (what does this mean?). If published, this will include your full peer review and any attached files.

Reviewer #1: Yes: Dipender Gill

Reviewer #2: No

---

## [Author Response · Author response to Decision Letter 0]

22 Aug 2019

Thank you so much indeed for these very helpful comments. We have responded all the reviewers' comments respectively and updated the manuscript and other related documents accordingly. Further details could be checked in the Response to Reviewers file and the Manuscript with Track Changes file.

---

## [Editor Report · Decision Letter 1]

23 Aug 2019

The effect of birth weight on body composition: Evidence from a birth cohort and a Mendelian randomization study

PONE-D-19-16847R1

Dear Dr. Schooling,

We are pleased to inform you that your manuscript has been judged scientifically suitable for publication and will be formally accepted for publication once it complies with all outstanding technical requirements.

With kind regards,

David Meyre

Academic Editor

PLOS ONE
---

## [Editor Report · Acceptance letter]

28 Aug 2019

PONE-D-19-16847R1 

The effect of birth weight on body composition: Evidence from a birth cohort and a Mendelian randomization study 

Dear Dr. Schooling:

I am pleased to inform you that your manuscript has been deemed suitable for publication in PLOS ONE. Congratulations! Your manuscript is now with our production department. 

With kind regards,

on behalf of

Dr David Meyre 

Academic Editor

PLOS ONE